# Therapeutic Management of Adults with Inflammatory Bowel Disease and Malignancies: A Clinical Challenge

**DOI:** 10.3390/cancers15020542

**Published:** 2023-01-16

**Authors:** Francesca Ferretti, Rosanna Cannatelli, Giovanni Maconi, Sandro Ardizzone

**Affiliations:** 1Gastrointestinal and Digestive Endoscopy Unit, ASST Fatebenefratelli Sacco, 20157 Milano, Italy; 2Department of Biomedical and Clinical Sciences “L. Sacco”, University of Milan, 20157 Milano, Italy

**Keywords:** inflammatory bowel disease, malignancy, neoplastic recurrence, IBD flare, immunosuppressants, azathioprine, biologic therapy, anti-TNFα, vedolizumab, ustekinumab, cancer progression

## Abstract

**Simple Summary:**

The management and treatment of patients with inflammatory bowel diseases after a diagnosis of malignancy represents a challenge for physicians, since their most common therapy, such as biologics and immunosuppressants, should be discontinued for 2–5 years after the end of cancer treatment. Special situations could be managed using new gut-selective drugs; however, limited data are available for these new therapies. We aim to summarize the current evidence about the reintroduction of different therapies after the primary diagnosis of cancer and to describe the course of inflammatory bowel disease without any immunosuppressive treatment after the diagnosis of cancer.

**Abstract:**

Patients with chronic inflammatory bowel diseases (IBD) have increased risk of developing intestinal and extraintestinal cancers. However, once a diagnosis of malignancy is made, the therapeutic management of Crohn’s disease (CD) and ulcerative colitis (UC) can be challenging as major guidelines suggest discontinuing the ongoing immunosuppressant and biological therapies for at least 2–5 years after the end of cancer treatment. Recently, new molecules such as vedolizumab and ustekinumab have been approved for IBD and limited data exist on the real risk of new or recurrent cancer in IBD patients with prior cancer, exposed to immunosuppressants and biologic agents. Thus, a multidisciplinary approach and case-by-case management is the preferred choice. The primary aim of our review was to summarize the current evidence about the safety of reintroducing an immunosuppressant or biologic agent in patients with a history of malignancy and to compare the different available therapies, including gut-selective agents. The secondary aim was to evaluate the clinical course of the IBD patients under cancer treatment who do not receive any specific immunosuppressant treatment after the diagnosis of cancer.

## 1. Introduction

Patients with chronic inflammatory bowel disease (IBD) have an increased risk of developing intestinal and extraintestinal cancers [1].

Whilst the chronic intestinal inflammation has been recognized as a pivotal cause of intestinal malignancies in an “inflammation-dysplasia-cancer” sequence, with a different mechanism from sporadic cancer [2], the association between extraintestinal cancers and IBD is more controversial. In 2010, a large meta-analysis [3] included 17,052 IBD patients and reported an overall risk of cancer “at any site” that was not significantly increased (SIR 1.10, 95% CI 0.96–1.27). However, a site-specific analysis demonstrated a higher risk of upper gastrointestinal tract, lung, urinary bladder, and skin cancers among Crohn’s disease (CD) patients, and liver-biliary cancer and leukemia in ulcerative colitis (UC) patients.

A combination of the chronic immune system dysregulation typical of inflammatory conditions [4] and the effects of immunosuppressive therapies could play a role in the onset of malignancies in these patients [3].

Major evidence applies to the incident cancer risk on immunosuppressed patients and it demonstrates an increased risk of lymphoma and nonmelanoma skin cancer (NMSC) in patients on treatments with thiopurines [5,6,7,8,9], while melanoma incidence is increased 1.32-fold in patients under TNFα inhibitors [10]. According to previous experience in organ transplant recipients, thiopurines, but not anti-TNF*α*, are associated with a higher risk of HPV-related cervical cancer [11]. Analogously, a higher rate of bladder and renal cancer was observed in immunosuppressed patients [12], with high risk of recurrence under treatment with thiopurines [13].

Considering that randomized controlled trials usually exclude neoplastic patients, it is expected that data about the effects of immunosuppressants in IBD patients with a prior or ongoing history of tumor will be very limited. However, given their potential role in the onset of incident cancers in immunosuppressed patients, major guidelines preventively suggest discontinuing the ongoing treatment with thiopurines and biologics for at least 2–5 years after the conclusion of cancer treatment due to the higher risk of recurrence in this period of time [10,14]. Moreover, their reintroduction after cancer treatment should be carefully evaluated both for the risk of local recurrence or metastatic spread as well as for the risk of developing a second malignancy.

This precautionary management can pose major challenges for gastroenterologists in the case of IBD flares. As conventional therapies are not associated with neoplastic evolution, guidelines support the use of aminosalicylates, nutritional approaches and corticosteroids in these cases [10]. However, it is known that in severely active IBD, this approach could be inadequate and a multidisciplinary discussion about the use of anti-TNF*α*, systemic corticosteroids, and/or surgery on a case-by-case manner is recommended.

In this field, evidence is mainly derived from other branches of medicine or limited real-world experiences. In particular, it has been reported that there is a higher risk of developing a new or recurrent cancer in transplanted patients, especially NMSC and lymphoma [12,15]. The first data on the effects of anti-TNF*α* therapy were drawn from patients with arthritis who seemed to have no increased risk to develop new or recurrent cancer after exposure to anti-TNF*α* compared to patients without anti-TNF*α* therapy [16].

About IBD patients, there is only one large prospective study promoted by the CESAME Group (Cancers Et Surrisque Associé aux Maladies inflammatoires intestinales En France). In this experiment, they enrolled 17,047 IBD patients and evaluated the risk of incident or recurrent cancer in 405 (2.4%) IBD patients with a personal history of cancer. Globally, a 1.9 multivariate-adjusted higher risk for incident cancer was observed in this cohort, independent from the use of immunosuppressant therapy [17,18].

Due to the growing prevalence of IBD all over the world and the well-known aging population, the therapeutic management of IBD patients with a newly diagnosed or a previous cancer is a challenge that both oncologists and gastroenterologists will increasingly face. Thus, up to now, if it is necessary to reintroduce or continue a biologic therapy, there are no strong and unique recommendations, but a multidisciplinary approach and a case-by-case management is usually the preferred choice for these patients. Besides the immunomodulators (IMM) such as azathioprine (AZA) and methotrexate (MTX) and the first biologic agents with anti-TNF*α* action, in the last few years, some new selective molecules have been introduced in the therapeutic scenario of IBD, such as vedolizumab (VDZ) and ustekinumab (USK). The different mechanisms of actions of biologic therapies are shown in Figure 1. The oncologic safety of these drugs relies on scanty real life retrospective data and quite short follow-up [19,20], but they could be a reasonable option in the future if their safety is confirmed.

This review aims to summarize the current evidence about the safety of continuing or reintroducing an immunosuppressive therapy (including both IMM and biologic agents) in patients with a history of malignancy, including in the research also gut-selective and recent molecules (Figure 1).

Moreover, as the stop of immunosuppressive therapy is correlated with a risk of relapse, a secondary aim of this review is to summarize the studies analyzing the clinical course of the IBD patients under cancer treatment who do not receive any specific immunosuppressant treatment after the diagnosis of cancer.

## 2. Materials and Methods

In order to collect the available studies dealing with this topic, we conducted a computerized search on PubMed and a manual literature search on October 2022, using the following query “(((azathioprine[Title/Abstract]) AND (cancer[Title/Abstract])) AND (malignancy[Title/Abstract])) AND (IBD[Title/Abstract])”. The term “azathioprine” was repeatedly replaced with all different biologics used in IBD management, such as “anti-TNF*α*” or “TNF*α* inhibitors” or “infliximab” or “adalimumab”, “vedolizumab”, “ustekinumab” and “tofacitinib”. To complete and fulfill the research, a general query with “biologics”/“immunosuppressants” and “cancer” was conducted. After a title and abstract screening, we excluded inconsistent or off-topic studies, editorials, experts’ opinions, or clinical cases. Finally, a full-text review of all remaining studies was performed, and we collected and organized the most relevant data.

## 3. Diagnosis of Cancer in IBD Patients: How to Manage IBD Therapies

The diagnosis of cancer in IBD patients is a major challenge both for gastroenterologists and oncologists.

As aforementioned, in the case of cancer diagnosis, the more recent IBD guidelines by ECCO [10] and BSG [14] agree on the need for discontinuing current immunosuppressive therapies. In particular, thiopurines, calcineurin inhibitors and anti-TNF*α* agents should be preventively discontinued during cancer therapy.

In these patients, in case of active disease, 5-ASA, nutritional therapies and local steroids are considered safe drugs. However, in severe flares, these options could be inadequate, and anti-TNF*α*, methotrexate, systemic corticosteroids, and/or surgery should be evaluated, and a multidisciplinary approach is recommended in these challenging cases [10].

In the case of skin cancers, thiopurines should be definitely stopped in patients with squamous cells or aggressive basal cell carcinomas or in case of multiple synchronous or sequential lesions. Instead, in nonaggressive basal cell carcinoma, they can be continued in the absence of other therapeutic options [10].

After stopping therapies, the reintroduction of immunosuppressants should be delayed for 2 to 5 years after the completion of cancer therapies, according to the risk of neoplastic recurrence [13]. In particular, a delay of 5 years is recommended in the case of breast, malignant melanoma, and renal cell carcinoma [14].

However, it is noteworthy that in a large meta-analysis by Shelton et al. [21] including 11,702 patients with immune-mediated diseases, they performed a stratified analysis focused on studies in which the median interval between the cancer treatment and immunosuppressants reintroduction was shorter than 6 years. This analysis did not show any statistical difference in terms of risk of new or recurrent cancers in this subset of patients. This result can be supportive in those selected cases in which an earlier reintroduction is advisable. In these cases, the decision should be discussed between oncologists and gastroenterologists, keeping in mind the risk and benefits of therapies and the patient’s quality of life. In particular, an early introduction of thiopurines or anti-TNF*α* only 3 months after the last chemotherapy with close oncologic follow-up has been attempted with a close monitoring by oncologists [22]. Moreover, as a general assumption, in these patients, a step-up approach starting with monotherapy should be preferred.

Moreover, despite the guidelines’ recommendation to discontinue the ongoing treatment, there are some studies evaluating patients who continued the immunosuppressive IBD therapies after cancer diagnosis.

Algaba et al. in 2015 [23] described the course of patients who continued IBD therapies despite a cancer diagnosis. In this multicenter observational study, 19/48 (39.6%) IBD patients maintained the immunosuppressive therapies after cancer diagnosis. In particular, thiopurines were maintained in about half of patients (13/27, 48.1%) in monotherapy, and one-third of patients (4/12, 33.3%) in combination with anti-TNF*α*. On the contrary, none of the six patients on anti-TNF*α* continued their therapies. By comparing the rate of cancer remission and death related to cancer in a 5-year follow-up between who continued the IBD treatment and who discontinued it, the results were comparable (95% vs. 74% and 5% vs. 8%, respectively). However, the authors remained cautious in suggesting the maintenance or early reintroduction of immunosuppressive therapy after a cancer diagnosis. 

Additionally, in the cohort of Axelrad et al. [24], about two-thirds of patients continued their IBD medications during cancer treatment and the decision was mainly related to the overall IBD course according to the indications of the gastroenterologist, but no specific analysis was performed.

In the study of Rajca [25], evaluating the impact of cancer diagnosis and treatment on IBD management, the diagnosis of cancer partially influenced the ongoing IBD therapy. Globally, in patients with a previous diagnosis of cancer, only 4/15 patients started AZA or MTX within the first three years of disease. However, in IBD patients with a new diagnosis of cancer, only 1/41 who were not already on immunosuppressants were subsequently placed on MTX, suggesting a sort of reluctance in starting immunosuppression in neoplastic patients. Among IBD patients already on immunosuppressants, AZA was stopped in 11/21 patients (and in 3, switched to anti-TNF*α*), while anti-TNF*α* was discontinued in 2 out of 3 cases. Globally, in this study, they observed a comparable use of anti-TNF*α* but a lower use of IMM (19 vs. 25%, *p* < 0.001) in patients with cancer vs. controls. Moreover, a higher rate of surgery was described (4 vs. 2.5%, *p* < 0.05).

Analogously, in the recent study by Vedamurthy et al. [26], they analyzed a cohort of IBD patients who reintroduced anti-TNF*α* or VDZ 1.3 to 3.9 years after cancer diagnosis and a subgroup of patients who did not interrupt therapy, and they did not find an increased risk of new or recurrent cancer in either the anti-TNF*α* or the VDZ group. An early introduction of USK (median time 12 months) has been reported by Hasan [27], and the analysis limited to patients treated with biologic in the first 4 years after cancer diagnosis did not find any difference between USK, VDZ, anti-TNF*α*, and controls. However, the large multicenter study by Poullenot et al. [28] showed that an interval < 12 months between the diagnosis of cancer and the start of immunosuppressants can be associated with a higher rate of incident cancer (HR 2.89, *p* = 0.01), independent from the type of immunosuppressants (including AZA, MTX, anti-TNF*α*, and VDZ). 

Thus, even though available evidence suggests that keeping the ongoing IBD therapy or the early reintroduction of an immunosuppressive monotherapy can be considered a safe choice, the limited and retrospective data did not support the authors in recommending this choice as a gold standard but an option to be evaluated case-by-case.

## 4. Risk of New or Recurrent Cancer in IBD Patients with a Previous History of Cancer Treated with Immunosuppressants

As mentioned in the introduction, the continuation or reintroduction of immunosuppressive therapies in IBD patients with a history of cancers is still controversial. The main results are drawn from studies of other branches of medicine and actual recommendations are weak. We collected the current results available on IBD patients and the employ of thiopurines (mainly AZA) and current biologic therapies, including novel and gut-selective molecules. Studies are summarized in Table 1. As shown, evidence is more consistent for AZA and anti-TNFα, due to their longer use in IBD patients; however, early but encouraging results can be observed for new agents such as VDZ and USK.

### 4.1. Thiopurines

Thiopurines act with a direct anti-inflammatory mechanism, inhibiting cytotoxic T-cells and natural killer cells [32]. They include azathioprine and 6-mercaptopurine (6-MP).

Even if, in the last decade, the introduction of different biologic options has reduced its employ, it can still be considered as a mainstay of maintenance therapy of IBD patients due to their definite steroid-sparing action and durable effects [10]. However, its employ is limited by different well-known adverse events and contraindications, among these, the association between a long-term use and an increased risk of neoplastic complications such as NMSC has been demonstrated [33]; conversely, its relationship with lymphoproliferative disorders is more controversial [34,35,36,37]. Immunosuppressed IBD patients who develop lymphoma are often EBV-positive, probably due to the cytotoxic effects of thiopurines on an activated immune system against EBV infection [38]. 

To date, the use of thiopurines in IBD patients with a previous history of cancer has been analyzed in different studies and evidence relies on a larger cohort of patients and longer follow-up compared to more recent drugs such as biologics. Of course, NMSC deserves separate indications. 

In the aforementioned large and prospective study by Beaugerie et al. in 2014 [18], 94/405 IBD patients with a personal history of cancer continued their immunosuppressants therapies. Of these, 77 were on thiopurines, 10 on methotrexate, 7 on TNFα inhibitors, and 4 on other immunosuppressants. Among them, 23 developed a new (16) or recurrent (7) cancer (21.1/1000 patient-years, p-y). Even if they did not demonstrate a significant difference in recurrent cancer rate in patients exposed or not exposed to any immunosuppressants, they noticed that only patients on AZA developed a recurrent cancer, while none on MTX, anti-TNF*α*, or other immunosuppressants developed a new or recurrent cancer. However, due to the limited number of IBD patients on anti-TNF*α*, low evidence could be drawn from this study. 

In order to overcome this limitation, the multicenter retrospective study of Axelrad et al. of 2016 [29] included 333 IBD patients, all with a previous history of cancer who were subsequently treated with immunosuppressants, including 22 antimetabolite (HR 1.08, 95% CI 0.54–2.15), 7 anti-TNFα (HR 0.32, 95% CI 0.09–1.09), and 15 anti-metabolite + anti-TNF*α* (HR 0.64, 95% CI 0.54–2.15). During the follow-up period (285–852 × 100-person years), 90/333 patients (27%) developed an incident cancer (44 new, 48 recurrent, and 8 new + recurrent). In this cohort, the exposure to immunosuppressant agents such as AZA or MTX was not associated with a significantly higher risk of incident cancer compared to other immunosuppressive drugs or non-immunosuppressed (controls) patients with an HR of 1.08.

In the recent systematic review and meta-analysis by Shelton et al. [21] including 11,702 persons with immune-mediated diseases contributing 31,258 person-years (p-y) of follow-up after a prior diagnosis of cancer, the pooled incidence rate of new or recurrent cancer in patients who received immunosuppressants was 33.8 per 1000 p-y, similar to patients who received anti-TNF or no immunosuppressants. In the subanalysis including only the IBD patients (3706 patients contributing 10,332 person-years of follow-up), the pooled incidence of new or recurrent cancer in patients receiving IMM was 37.9 per 1000 p-y, without statistical difference with the other groups of treatments (no immunosuppressants or anti-TNF*α*). However, in two studies including index skin cancers alone, the risk of new or recurrent cancers was statistically higher in the group on IMM (71.6 per 1000 p-y) compared to patients without therapy (50.8 per 1000 p-y, *p* = 0.035) and numerically higher than anti-TNF*α* (55.5 per 1000 p-y, *p* = ns).

In conclusion, these studies agree and confirm the nonsignificant effect of thiopurines on new or recurrent cancer risk compared to other therapies or controls, differently from transplant studies in which thiopurines appear to be associated with an overall increased risk of malignancies [12].

Regarding NMSC, available studies already demonstrated an association between thiopurines employed not only with new cases of NMSC [7,39], but also with cancer recurrence. In particular, Scott et al. showed an increased risk (nearly statistically significant) of repeated occurrence of squamous cell carcinoma (SCC) and basal cell carcinoma (BCC) with a longer duration of thiopurines treatment (>1 year) after the NMSC diagnosis, not before [40]. Moreover, in case of previous SCC in IBD patients, Khan et al. showed that continuing thiopurines was associated with a higher risk of SCC-related mortality [41]. More recently, according to a large retrospective observational study by Khan et al. [42] on 518 IBD patients with a history of BCC, an increased risk of about 1.65 times for recurrence of cancer was observed in patients on active/continued treatment with thiopurines (adjusted HR 1.65, 95% CI 1.24–2.19, *p* < 0.001), compared with patients on 5-ASA (12.8 × 100 person-years). It is noteworthy that this risk was no longer present 6 months after the discontinuation of thiopurines and/or switching to anti-TNF*α* (an adjusted HR 1.22 (95% CI 0.86–1.74, P 5 0.26)), confirming a disbalance between the risk and benefits of continuing AZA after a history of BCC. In conclusion, in case of the conscious decision of continuing the therapy, frequent dermatological screening and a timely removal of recurring skin lesions are recommended.

### 4.2. Anti-TNFα

The TNFα inhibitors include both intravenous and subcutaneous formulations such as infliximab (IFX), adalimumab (ADA), and golimumab (GOL). These drugs bind and block both soluble and transmembrane TNFα receptors. 

Their role in induction and the maintenance of remission in both UC and CD has been largely demonstrated [43,44], as well as their good safety profile, as their use is not associated with severe infections or an overall higher rate of malignancies [45,46]. However, a slightly higher risk of melanoma incidence was observed [8], with an incidence rate ratio (IRR) of 1.29 and odds ratio (OR) of 1.88 (higher for CD 1.95 than UC 1.73) [47]. In a study on a huge Sweden cohort describing a small absolute risk of melanoma, patients under anti-TNFα had an increased relative risk of melanoma of 50% compared to patients without anti-TNFα [48]. Although other studies have not confirmed this described association [49,50], a periodic dermatologic screening is still recommended [10]. Once the diagnosis of melanoma in IBD patients under anti-TNF*α* therapy is made, different options of management could be taken into account. Firstly, in patients with stage I of melanoma and active IBD or high risk of relapse, the choice to continue the anti-TNF*α* could be considered, alongside a close dermatological follow-up. Secondly, IBD patients with clinic, endoscopic, and/or histological remission could discontinue biological therapy, with 50% risk of relapse. Finally, another potential option is to change biological therapy with different molecules, in particular the new gut-selective VDZ. Surgery should be carefully discussed and reserved for patients with multiple risk factors for melanoma, poorly controlled IBD, or poor compliance to follow-up [51]. 

According to the multicenter retrospective study of Axelrad et al. of 2016 [29], the exposure to anti-TNF*α* agents or a combination of anti-TNF*α* and AZA were not associated with increased risk of incident cancer compared to other immunosuppressive drugs or non-immunosuppressed patients (HR 0.32 and 0.64, respectively). It is noteworthy that the duration of anti-TNF*α* treatment did not seem to affect the risk of incident cancer or its type and its recurrence risk. Moreover, at 5 years, no significant difference in cancer free survival was observed between immunosuppressed and non-immunosuppressed patients. 

Poullenot et al. [30] analyzed the risk of incident new or recurrent cancer in a multicenter cohort of 79 IBD patients with a recent history of cancer (median time 17 months, range 1–65; mainly breast and skin cancers) who were subsequently treated with at least one anti-TNF*α* (53 IFX, 26 ADA). The overall crude incidence rate of cancer was 84.5% per 1000 p-y (95% CI 83.1–85.8) with 15 patients (19%) developing recurrent cancers (8) and new cancers (7), including 5 BCC, in a median follow-up of 21 months (1–119). Survival without incident cancer was 96%, 86%, and 66% at 1, 2, and 5 years, respectively. 

These first studies were controversial: the results of Axelrad et al. [29] are in line with previous rheumatologic studies, demonstrating no difference in terms of new or recurrent cancer among patients on anti-TNF*α* and DMARDs [16,52]. Conversely, Poullenot observed a crude incidence rate in IBD patients (84.5%) that was significantly higher than in reported rates for the rheumatoid arthritis (RA) population, ranging from 25.3 to 45.5 per 1000 p-y [16,52]. Thus, data from Poullenot et al. seem to be less reassuring, even if they are cautious in interpreting their results identifying a substantial difference between IBD and RA populations as RA patients are older and more frequently treated with MTX-containing (and not AZA-containing) regimens [30].

In the same year, the aforementioned meta-analysis by Shelton et al. [21] supported Axelrad’s results [29]. In the subanalysis including only IBD patients (3706 patients contributing 10,332 person-years of follow-up), the pooled incidence of new or recurrent cancer in anti-TNF*α* was 48.5 per 1000 p-y without statistical difference with the other group of patients, even if the value was higher compared to the other groups (no therapy 35.7 per 1000 p-y and IMM 37.9 per 1000 p-y, *p* > 0.30).

Similarly, a larger and more recent study from Vedamurthy et al. [26] included 184 IBD patients with a previous cancer diagnosis treated with anti-TNFα. Out of these, 77 patients never interrupted the anti-TNF*α* therapy after the diagnosis of the index cancer and 30% were also exposed to thiopurines or methotrexate after the diagnosis of the index cancer. The median time to start anti-TNF*α* was 1.3 years (0–38 years). In the follow-up, there were 61 incident cancers (27 new, 34 recurrences) with a yielding incidence rate of 4.2 per 1000 p-y. In the multivariate analysis, neither anti-TNF*α* nor anti-TNF*α* + IMM therapy was associated with increased risk of cancer recurrence or new cancer development (HR 1.03, 95% CI 0.65–1.64 and HR 1.22; 95% CI 0.75–1.99, respectively) compared to patients without any immunosuppressant therapy. The results were the same in the restricted analysis of the biologic initiation within the first 5 years after the diagnosis of the index cancer (HR 0.68, 95% CI 0.37–1.22) and in the subanalysis where NMSCs were excluded (HR 0.87, 95% CI 0.45–1.65).

As concerns NMSC, in the analysis by Khan et al. limited to BCC, neither anti-TNF*α* alone (an adjusted HR 1.27 (95% CI 0.84–1.90, P 5 0.26)) nor anti-TNF*α* in combination with thiopurines (adjusted HR 1.37 (95% CI 0.90–2.08, P 5 0.14)) was associated with a higher risk of BCC recurrence compared with 5-ASA [42].

In conclusion, in the majority of the studies, anti-TNF*α* therapy seems to not increase the rate of new or recurrent diagnosis of cancer in patients with previous diagnosis of cancer. However, the decision should be made case-by-case and the combo therapy should be avoided.

### 4.3. Vedolizumab

VDZ is an anti-integrin which binds the α4β7 integrin and interferes with the gastrointestinal homing of T lymphocytes approved for the induction and maintenance therapy in IBD [53,54]. The long-term analysis of 8-year-follow-up confirmed the safety profile of the drug, especially regarding infections and malignancies [20]. 

Its approval for UC and CD is more recent compared to anti-TNF*α*, thus the available evidence on neoplastic patients is scarcer. However, it is noteworthy that due to its mechanism of action it is considered a “gut-selective” agent. This characteristic makes it a preferential option for elderly and comorbid patients. Moreover, its safety in terms of infection rates or risk of malignancies has already been confirmed in both pivotal clinical trials and real-life studies [55,56,57,58]. 

In the retrospective study by Vedamurthy et al. [26], 96 patients were exposed to VDZ after the diagnosis of index cancer with a median time of 3.9 years (range 0.1–43 years) and 31% were also exposed to thiopurines or MTX. In the VDZ group of patients, there were 18 (7 new cancer, 11 recurrences) events with a yielding incidence rate of 2.2 per 1000 p-y. In the multivariate analysis, therapy with VDZ was not associated with an increased risk of cancer recurrence or new cancer development (HR 0.72, 95% CI 0.38–1.36) compared with patients who did not receive any treatment, even in the restricted analysis to biologic initiation within the first five years after the diagnosis of the index cancer (HR 1.10, 95% CI 0.57–2.12). The same results were reached in the subanalysis excluding patients with NMSC (HR 0.56, 95% CI 0.23–1.39).

More recently, a retrospective study [31] enrolled 390 IBD patients with a diagnosis of cancer after diagnosis of IBD who received medical care in an academic medical center between 2013 and 2020. Amongst them, 37 were exposed to VDZ, 14 USK, 41 anti-TNF*α*, and 31 IMM, and 267 were not under treatment. A total of 79 patients (20%) developed subsequent cancer during the follow-up of 52 months. Amongst them, 61 (15.6%) were new cancer and 18 (4.6%) were recurrent cancer. The subsequent cancer rate per 100 p-y for VDZ was 1.9 (6 patients amongst 37 exposed to VDZ), and it was not statistically different from patients exposed to USK, anti-TNF*α*, IMM, or no therapy (3.0, 3.8, 1.6, and 2.7, respectively, *p* = 0.41). Moreover, patients exposed to more than one biologic during the follow-up did not increase the risk of subsequent cancer. Even in the subanalysis excluding either NMSC or gastrointestinal malignancies, the adjusted risk for subsequent cancer did not change. 

A multicenter French cohort study [28] included 538 IBD patients with a confirmed diagnosis of cancer from 33 centers from May 2016 to December 2019, with a median follow-up of 55 months. In this period, 231 (43%) of patients did not receive any immunosuppressants, whilst 307 (57%) received IMM or biologic therapies with a median interval of 12 (0–45) months after cancer diagnosis: specifically, 112 (21%) anti-TNF*α*, 64 MTX (12%), 79 (15%) thiopurines, 48 (9%) VDZ, and 4 (1%) USK (excluded from the analysis due to the limited number of patients). There were 100 (19%) incident cancers, 62 (12%) recurrences, and 38 (7%) new cancers. There was no statistical difference in the adjusted (after matching on age, lymph nodes, metastasis, and Penn’s classification) cancer incidence rates per 1000 p-y in patients under anti-TNF*α*, VDZ, and no treatment (41.4, 33.6, and 40.9, respectively). In the multivariate analysis on 307 patients who received immunosuppressive treatments, the type of treatment was not associated with the risk of incident cancer (*p* > 0.9). However, a time < 12 months from the diagnosis of cancer and the start of immunosuppression was associated with an increased risk of incident cancer (HR 2.89).

In conclusion, to date, in all the published studies about VDZ in patients with a previous diagnosis of cancer, the safety of the drug was confirmed in both new and recurrent cancer.

### 4.4. Ustekinumab

USK is an antagonist of the p40 subunit of interleukin-12 and interleukin-23 approved for the treatment of either UC or CD, which has shown a good safety profile [59,60]. In the 5-year-follow-up, USK was not associated with an increase of malignancies [19]. Due to the recent adoption of this drug in IBD, only a few studies have been available about the use of USK in patients with a previous diagnosis of cancer. 

In a recent retrospective study by Hong et al. [31] including 390 IBD patients with a diagnosis of cancer after diagnosis of IBD, the subsequent cancer rate per 100 p-y for USK was 3.0 (2 patients amongst 14 exposed to USK), and it was not statistically different from patients exposed to VDZ, anti-TNF*α*, IMM, or no therapy (*p* = 0.41). In the multivariate Cox model adjusting for age at index cancer, IBD type, smoking, cancer recurrence risk, cancer stage, time from cancer to treatment, neither VDZ nor USK was associated with the subsequent cancer development (adjusted hazard ratio 1.36 and 0.96, respectively).

In the retrospective study from Hasan et al. [27], they included 341 IBD patients from Cleveland Clinic from 2014 to 2020. Amongst them, 160 were exposed to biologics (34 VDZ, 27 USK, and 99 anti-TNF*α*) after the diagnosis of cancer and 181 received no immunosuppressive treatment. There were 27 subsequent cancers (18 new and 9 recurrence) in the control group with an incidence rate of 2.4 per 100 p-y and only 1 recurrent cancer in patients under biologics, in particular, anti-TNF*α*, with incident rates for cancer per 100 p-y of 0.4 for VDZ, 1.8 for USK, and 0.7 for anti-TNF*α*. In the multivariate analysis including IBD type, age at the diagnosis of index cancer, stage of cancer, premalignancy treatment with systemic steroids, radiation, chemotherapy, and surgical treatment for primary cancer, neither USK (HR 0.88, *p* = 0.833) nor VDZ (HR 0.18, *p* = 0.096) nor anti-TNF*α* (HR 0.47, *p* = 0.087) were found to increase the risk of incident cancer. Similar results were found in the subanalysis excluding NMSC or in the subanalysis limited to patients who had received biologics within the first 4 years after the index cancer.

Similar to VDZ, the available data for USK was comforting in both new or recurrence cancer, reinforcing the safety profile of the drug.

### 4.5. Novel Therapies

Tofacitinib is a new therapeutic option available for UC, and it is an oral, small-molecule Janus kinase inhibitor [61]. Due to its extremely recent introduction as a therapeutic option in moderate-to-severe UC, there are no available studies on tofacitinib and its real-life effects in patients with previous neoplastic conditions. However, in the OCTAVE trial [61], there was no diagnosis of malignancies, except NMSC, in the induction trial and only one breast cancer in the sustained trial in a patient who received placebo. In the long-term extension study, there were 20 diagnoses of malignancies, except NMSC, on 1124 UC patients, without any apparent clustering of malignancies [62].

Other new therapies are going to be available for the treatment of IBD in the near future. In particular, ozanimod [63], a sphingosine-1-phosphate (S1P) receptor modulator, and upadacitinib [64], an oral selective inhibitor of Janus kinase 1, are promising options but evidence is too limited. 

## 5. Risk of IBD Relapse in Patients with Cancer Who Discontinue Immunosuppressants

A worrisome risk of discontinuing the maintenance IBD treatment in patients with a new diagnosis of cancer is a flare of the chronic disease. Sometimes gastrointestinal symptoms could be managed by an increase of the mesalamine dosage or a short-term use of corticosteroids; in more severe situations, surgery is a potential option. Few studies explored the risk of relapse during the cancer treatment in patients who discontinued immunosuppressant or biological therapy.

In 2012, Axelrad et al. [24] evaluated a cohort of 84 IBD patients with solid extraintestinal neoplasia with a follow-up observation time of 18 years. In this cohort, 15 patients had active IBD (symptomatic disease and/or active disease on endoscopy with histologic confirmation) at cancer diagnosis, two-thirds of those (10/15, 66.7%) showed remission of disease during cancer treatment (cytotoxic chemotherapy, hormonal therapy, or a combination of both), while the other 5 patients with active disease received only hormonal therapy. Conversely, 69 patients were on remission at cancer diagnosis, of those, about 17% (12/69) showed an IBD flare.

The main predictive factor of IBD course was the ongoing therapy for cancer, as 90% of patients on cytotoxic chemotherapy maintained IBD remission compared to 64% of patients under a combination of cytotoxic chemotherapy and adjuvant hormone therapy (HR of IBD flare 12.25; 95% CI, 1.51–99.06) or assuming only the hormone therapy (HR of IBD flare 11.56; 95% CI 1.39–96.43) with a *p* = 0.02. Other factors such as age at IBD diagnosis (HR, 0.96; 95% CI, 0.92–1.01), a diagnosis of CD (HR 3, 95% CI, 0.96–9.42), and exposure to infliximab (HR, 2.39; 95% CI, 0.42–13.75) were associated with an increased risk of active IBD during bivariate but not multivariable analysis. Of these, 43/69 (62.3%) of inactive IBD and 10/14 (66.7%) of active IBD continued through cancer treatment, but this variable was not statistically significant in bivariate analysis (HR 0.79, 95% CI 0.29–2.18).

Rajca et al. [25] evaluated (2014) the impact of extraintestinal cancer diagnosis on 80 IBD patients (33 M, median age at cancer diagnosis: 48 years). Compared to a matched population of controls, 29 patients on chemotherapy did not show a significantly different IBD activity both in the short- and mid-term. Of them, 11 patients (38%) developed a flare during the year of chemotherapy, while IBD was active in up to 49% of patients who did not receive chemotherapy. A similar effect was observed in the two following years with about one-third of patients experiencing IBD flare both with or without chemotherapy (33% active disease on chemotherapy versus 32% without chemotherapy).

In 2015, Algaba et al. [23] showed that 6 patients had an IBD flare in the follow-up, of those, 4 were not receiving IBD therapy, 1 AZA, and 1 ADA: they required adding thiopurines and/or anti-TNF-α drug treatment (2 ADA, 2 AZA, 1 AZA + IFX, and 1 MTX + ADA). Cancer remission was observed in all cases of reintroduction.

Thus, according to the few available studies, there is a risk of IBD flare in patients discontinuing therapy, but it is not significantly different from controls. The aforementioned parameters such as age, CD diagnosis, and use of infliximab which seem to relate to higher risk of IBD flare can be explained by their already known association with a more severe course of IBD also in non-neoplastic patients. 

Moreover, Axelrad et al. suggest that active IBD can benefit from cytotoxic chemotherapy, that the risk of flare within 5 years is about 17% and that hormonal therapies may increase the risk of IBD recurrence. The beneficial effect of cytotoxic drugs may be due to cell death and avoiding cell division as T lymphocytes with an immunosuppressive effect. On the contrary, hormonal therapies seems to relate to IBD flares; indeed, there is some evidence about the interaction between hormonal status (hormonal contraception, pregnancy, or hormone replacement therapy) and the risk of developing IBD and disease activity. However, this relationship is still controversial and further studies are necessary to confirm this potential causative effect on IBD flares. 

In recent years, the use of immunotherapy could also be successful in some refractory cases. These are particular and challenging patients due to the immune stimulation that could cause immune-related colitis similar to IBD as clinical symptoms, endoscopic appearance, and pathological infiltrate, with minor changes [65]. In particular, patients under anti-cytotoxic T lymphocyte antigen, such as ipilimumab, or antiprogrammed death-1, such as nivolumab, pembrolizumab, and atezolizumab, have been found linked to the development of IBD-like colitis, more common in patients under more than one therapy. However, this colitis responded to the same treatment for IBD, in particular, steroids and anti-TNF*α* [66,67,68,69].

Similarly, patients with an established diagnosis of autoimmune disease who received ipilimumab could experience an immune-related toxic effect. A recent multicenter retrospective study [70] involving 30 patients with autoimmune disease, amongst which, 6 had IBD. Two IBD patients, one CD and one UC, experienced an immune-mediated adverse event (colitis) and a clinical relapse of the disease, respectively. The patient with CD responded to treatment with systemic steroids, whilst the patient with UC, with prior colectomy, had a partial response to dexamethasone and IFX. Therefore, the authors concluded that, despite 50% of patients with autoimmune diseases experiencing an exacerbation of the disease after treatment with ipilimumab, these events were successfully treated by standard therapy and should not preclude the use of the drug.

Moreover, some studies have investigated the effect of IBD treatment on cancer courses. Anti-TNF*α* seemed to increase cachexia and chemotherapy tolerance in patients with nonsmall lung cancer, renal cell carcinoma, and pancreatic cancer [71,72,73]. Moreover, patients with RA who experienced a diagnosis of cancer during treatment with anti-TNF*α* have not shown a worse prognosis of the underlying cancer [74].

To summarize, in most cases, cytotoxic chemotherapy works on the cancer and contributes to maintaining IBD remission, whilst the hormonal and immunotherapy could cause a relapse of the disease. However, sometimes, the relapse of the disease could require biological therapy, despite the use of chemotherapy.

## 6. Discussion and Future Perspectives

Due to the lack of large prospective studies or RCT with long-term results, actual evidence (Table 1) on patients with a newly diagnosed cancer or a prior history of malignancy is inadequate to outline a definite algorithm. Thus, both international guidelines [10,14] and more recent experts’ opinions [75] recommend a multidisciplinary effort to manage these patients.

A balance between the risk of cancer progression or recurrence and the IBD patients’ quality of life and intestinal health should be the therapeutic target for both oncologists and gastroenterologists (Figure 2). From the gastroenterologist’s point of view, a thorough assessment of IBD activity and risks of disease progression should be performed including clinical, laboratory, and objective evaluation of disease activity (clinical scores, laboratory tests including hemoglobin levels, C-reactive protein, fecal calprotectin, colonoscopy, and/or gastrointestinal imaging). Moreover, it is essential to also include in this evaluation patients’ quality of life and psychological measures. On the other hand, the oncologists should evaluate the type of new or previous cancer, the clinical history of patients, and the risk of recurrence according to Penn’s classification [13].

Finally, in case of the need for reintroducing an immunosuppressant, the medical staff should consider different parameters, including: the type of previous cancer: for example, in case of cervical or anal carcinoma, HPV-related thiopurines are forbidden, as well as in case of EBV-related lymphoproliferative disorder or NMSC [7,8,37,76,77,78];the ongoing immunosuppressants, if there was one, at the moment of previous cancer onset (i.e., melanoma under anti-TNF), in order to opt for another class of drug [51];the risk of cancer progression: in case of intermediated or high risk of recurrence, monotherapy with IMM or anti-TNF or VDZ should be adopted. Combo therapy should be used only in special and well-discussed situations.

However, the recent introduction of drugs with a gut-selective action has opened up new horizons in the management of these complex patients, requiring a continuous and progressive update of our decisions. A recent review suggested to employ VDZ and USK in most patients with a history of malignancy [75], even if, as shown in the aforementioned literature review, evidence on novel drugs is very limited and further studies are needed.

The active IBD itself is considered a risk factor for the malignancies [4,79]; thus, in patients with high risk of relapse or active disease, the activity of the disease should be balanced with the risk of cancer recurrence (Figure 2). A multidisciplinary and in-depth evaluation of the patients is crucial to adopt a tailored decision, making them aware of their condition and the type of treatment proposed. A special mention to patients with an active tumor: these patients should potentially discontinue their immunosuppressive therapies, in consideration of the protective role of cytotoxic chemotherapy against IBD flares.

## 7. Conclusions

In conclusion, IBD patients with a cancer diagnosis still represent a challenge for both gastroenterologists and oncologists. The correct evaluation of the intestinal disease and cancer (Penn’s classification) is pivotal in the decision of the tailored therapy. The advent of new gut-selective drugs could help the decision to switch the type of biological therapy. However, a multidisciplinary approach involving different specialists is still to be preferred, especially in tertiary referral centers. Further prospective and more focalized studies are needed to guide clinicians in adopting adequate patient-centered decisions.

## Figures and Tables

**Figure 1 cancers-15-00542-f001:**
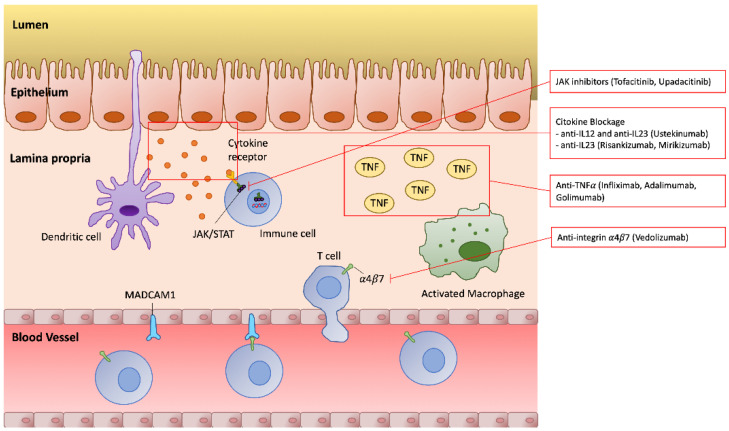
Mechanisms of action and targets of current biologic therapies in IBD.

**Figure 2 cancers-15-00542-f002:**
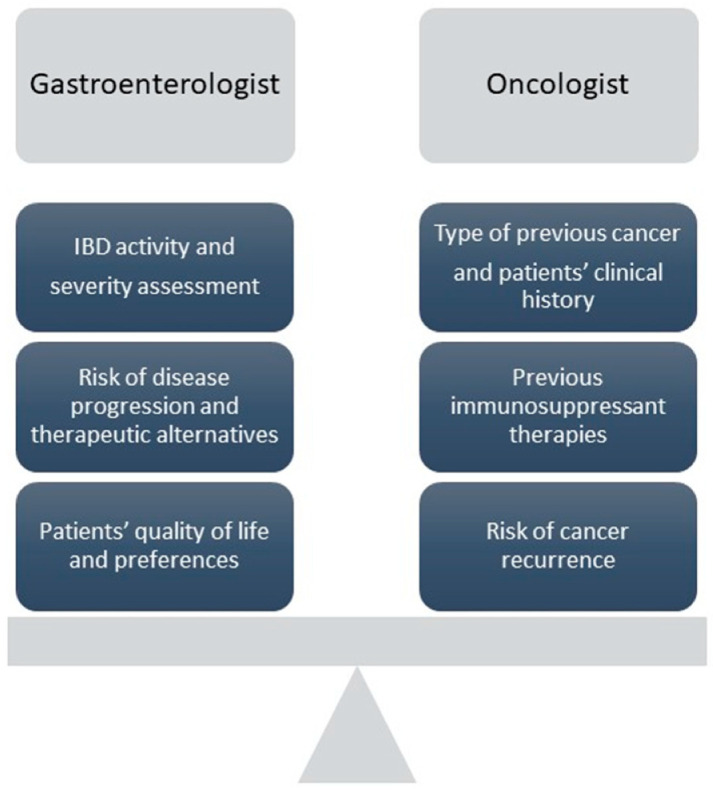
Balanced evaluation of clinical factors to be considered by gastroenterologist and oncologist.

**Table 1 cancers-15-00542-t001:** Studies on IBD patients with previous diagnosis of cancer under immunomodulators or biologic therapies.

Study	Year	Type of Study	IBD Patients (*n*)	Therapies	Follow-Up	Results
Beaugerie et al. [18]	2014	Prospective	405	91 IMM 7 anti-TNF	Median 36 months (IQR 32–43) 1091 p-y	There was no significant difference in the incidence of new or recurrent cancers between patients exposed or not exposed to any immunosuppressant.
Axelrad et al. [29]	2016	Retrospective	333	55 anti-TNF 51 anti-TNF + IMM 78 IMM 149 no IS	NA	No statistical difference in type (*p* = 0.61) or time (*p* = 0.14) of incident cancer. No statistical difference in risk of incident cancer or time of subsequent cancer (*p* = 0.22) between 4 groups.
Poullenot et al. [30]	2016	Retrospective and prospective	79	61 anti-TNF 18 anti-TNF+IMM	Median 21 months (1–119)	Crude incidence rate of cancer was 84.5 per 1000 p-y. It was 74.8 per 1000 p-y in pts with IMM and 87.3 per 1000 p-y in pts without IMM.
Shelton et al. [21]	2016	Systematic review and meta-analysis	3706	N/A	10,332 p-y	There was no statistical difference in pooled incidence rate per 1000 p-y between anti-TNF (48.5), IMM (37.9), and none (35.7), *p* > 0.30.
Vedamurthy et al. [26]	2020	Retrospective	463	184 anti-TNF 96 VDZ 183 no IS	Median 6.2 p-y	No increase in the risk of new or recurrent cancer in pts under anti-TNF (HR 1.03) and VDZ (HR 1.38) compared to pts without IS treatment, after adjusting for confounders.
Hasan et al. [27]	2022	Retrospective	341	99 anti-TNF 34 VDZ 27 USK 181 no IS	Median 5.2 p-y	No increased risk of incident cancer in patients receiving USK (HR 0.88), VDZ (HR 0.18), or anti-TNF (HR 0.47).
Hong et al. [31]	2022	Retrospective	390	41 anti-TNF 37 VDZ 14 USK 31 IMM 267 no IS	Median 52 months	No increase in subsequent cancer with VDZ (adjusted HR 1.36) or USK (adjusted HR 0.96). No increased risk in subsequent cancer in patients with more than 1 biologic exposure.
Poullenot et al. [28]	2022	Retrospective	538	112 anti-TNF 48 VDZ 4 USK (excluded) 143 IMM 231 no IS	Median 55 months (IQR 23–100)	Crude incidence rates for cancer were not different in patients under anti-TNF (36.6), VDZ (33.6), or no treatment (47.0), *p* = 0.23.

anti-TNF: antitumor necrosis factor; HR: hazard ratio; IMM: immunomodulators (azathioprine or methotrexate); IS: immunosuppressants; IQR: interquartile range; N/A: not applicable; USK: ustekinumab; VDZ: vedolizumab.

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
