# Peer review of "Therapeutic Management of Adults with Inflammatory Bowel Disease and Malignancies: A Clinical Challenge"

_cancers, 2023, doi:10.3390/cancers15020542_

Round 1

Reviewer 1 Report

This short review by Francesca Ferretti, et al. summarizes the current evidences about the re-introduction of different therapies after the primary diagnosis of cancer and report the course of IBD without any immunosuppressive treatment after the diagnosis of cancer. IBD patients have increased risk of developing intestinal and extra-intestinal cancers. However, when they are diagnosed malignancy, the therapeutic management of IBD can be challenging as major guidelines suggest discontinuing the ongoing immunosuppressant and biological therapies for at least 2-5 years after the end of cancer treatment. Recently, several new molecules, including Vedolizumab and Ustekinumab, approved for IBD have limited data on the exact risk of new or recurrent cancer in IBD patients with prior cancer, exposed to immunosuppressants and biologic agents. The authors concluded that a multidisciplinary approach and case-by-case management is the preferred choice.

This reviewer is slightly hard to follow the text, but this is an important review in this research field. I would recommend using figure(s) and/or tables, where introduce mechanisms of action of new molecules for easy understanding the contents.

Reviewer 2 Report

This manuscript represents a very much needed review on IBD patients diagnosed with cancer, extremely precious for our practice, both GIs and oncologists, a real gem. Aims of this review were very generous and were successfully achieved. I truly appreciate the Authors’ idea, the conceptualization, the design and the in-depth analysis and synthesis. Hard work, but it was worth it. Also, “Discussion and future perspectives” paragraph is of great value. Congratulations to the Authors! Given the enormous data the Authors had to include, there were some inconsistencies, that I mention, along with minor comments/suggestions. Thank you.

1. Title: Maybe the Authors would consider changing the title to: “Malignancies in adults with inflammatory bowel disease: a challenge for the therapeutic management”, as the manuscript refers to therapeutic options after the primary diagnosis of cancer and also the course of IBD. From the present title, I thought of something else - like IBD and risk of malignancies – a challenge for the therapeutic management. Also, the paper presents scientific evidence in adult patients.

2. Simple Summary and Abstract: line 11 and line 21: maybe replace “since” with “after” the end.

3. Lines 20- 21: the Authors wrote “major guidelines suggest discontinuing the ongoing immunosuppressant and biological therapies for at least 2-5 years since the end of cancer treatment”, like in Simple Summary. But, in “Introduction” – lines 61-62 – the Authors wrote “major guidelines suggest discontinuing the ongoing treatment with thiopurines and biologics for at least 2-5 years after the diagnosis of cancer”. Please decide and revise. It is not the same. Also, please include this in lines 131-132.

4. “Introduction: Line 57: Please revise “Considering these current recommendations”. There were no mentioned recommendations written before.

5. L 73: please delete “was observed”. Please define “NMSC” before its abbreviation.

6. L 92: please correct “should have been”.

7. L 96-96: I think the formulation of the Aim should not include “in terms of new or recurrent cancers” and just what is written in the Abstract: “in patients with a history of malignancy”. Or please rephrase, as it appears confusing (long sentence).

8. L 105: Please write “PubMed”.

9. Please mention also whether the search included 6-mercaptopurine (not only azathioprine), methotrexate, cyclosporine, and tacrolimus. You mentioned immunomodulators, but it would not hurt to mention them specifically.

10. L 112-114: Please correct the sentence, in terms of the English language: “The full-text of any included studies were thoroughly reviewed in order to identify the more relevant and collect their results. “

11. L 118: Please replace “more” with “most”.

12. L 142: Please define “IMM” before its abbreviation.

13. L 139-144: Please shorten this long sentence, in order to be clearer.

14. L 152: Please replace “if” with “in”.

15. L 173: please insert “vs controls” after “in patients with cancer”.

16. L 176: I would suggest mentioning clearly, instead of” immunosuppression”, Vedolizumab or anti-TNFα, as these two were studied. 

17. L 178: Please define abbreviation “VDZ” before its first use.

18. L 179: Please delete “also”.

19. This is very important: L 185: IMM was not defined. I thought it referred to Immunomodulators, as in line 142, it was written: “IMM or anti-TNF”. However, in L 185, anti-TNF alpha are considered part of IMM, as it is VDZ. Please revise. Page 234 – IMM are indeed defined as immunomodulators. Line 356 mentions IMM, VDZ, and anti-TNFα separately. Line 367-370: “In this period, 231 (43%) of patients did not receive any IMM treatments; whilst 307 (57%) received any biologics with a median interval of 12 (0-45) months after cancer diagnosis; in particular, 112 (21%) anti-TNFα, 79 (15%) thiopurines, 48 (9%) VDZ. Please revise, as it appears confusing: 307 any biologic – but 70 had thiopurines. Then, in Line 375, it is written “307 patients who received immunomodulator treatments”. Table 1 – title: “Studies of IBD patients with previous diagnosis of cancer under immunosuppressants or biological therapies”. Here, immunosuppressants are considered separately from biologics, even though in the manuscript the authors mostly considered that immunosuppressants included biologics. Table 1 legend: IMM are considered azathioprine or methotrexate. Please carefully revise throughout the whole manuscript – what is defined as immunosuppressants, what as immunomodulator, what is really IMM and so on.

20. L 219: please correct the year – 2016, as it appears in line 281 and reference.

21. L 246: Please define “SCC and BCC” before their abbreviation.

22. “Subparagraph 4.1 Azathioprine” should not be about thiopurines (including 6-MP) ?

23. L 333 – please replace “are” with “is”.

24. Lines 365-367 – please include the verb. Same for Lines 395-396.

25. Table 1 – I suggest to be inserted in the text much earlier, maybe in the Subparagraph 4, so that the readers are aware of its existence, not only when reading Discussion. Also, please correct “none” to “no” (last study by Poullenot). 

26. Please, after defining an abbreviation, do use it, without defining it again or without writing again the entire name [e.g. after VDZ was used many times before, subparagraph 4.3 mentions Vedolizumab (VDZ), same for subparagraph 4.4 - Ustekinumab (USK) etc etc].

Round 2

Reviewer 1 Report

The revised manuscript has greatly been improved.